# Systematic examination of preprint platforms for use in the medical and biomedical sciences setting

Jamie J Kirkham ,[1] Naomi C Penfold ,[2] Fiona Murphy,[3] Isabelle Boutron,[4] John P Ioannidis,[5] Jessica Polka,[2] David Moher [6]

► Prepublication history and supplemental material for this paper is available online. To view these files, please visit the journal online (http://dx.doi.org/10.1136/bmjopen-2020-041849).

For numbered affiliations see end of article.

**Correspondence to**
Professor Jamie J Kirkham; jamie.kirkham@manchester.ac.uk

## ABSTRACT

**Objectives** The objective of this review is to identify all preprint platforms with biomedical and medical scope and to compare and contrast the key characteristics and policies of these platforms.

**Study design and setting** Preprint platforms that were launched up to 25 June 2019 and have a biomedical and medical scope according to MEDLINE's journal selection criteria were identified using existing lists, web-based searches and the expertise of both academic and non-academic publication scientists. A data extraction form was developed, pilot tested and used to collect data from each preprint platform's webpage(s).

**Results** A total of 44 preprint platforms were identified as having biomedical and medical scope, 17 (39%) were hosted by the Open Science Framework preprint infrastructure, 6 (14%) were provided by F1000 Research (the Open Research Central infrastructure) and 21 (48%) were other independent preprint platforms. Preprint platforms were either owned by non-profit academic groups, scientific societies or funding organisations (n=28; 64%), owned/partly owned by for-profit publishers or companies (n=14; 32%) or owned by individuals/small communities (n=2; 5%). Twenty-four (55%) preprint platforms accepted content from all scientific fields although some of these had restrictions relating to funding source, geographical region or an affiliated journal's remit. Thirty-three (75%) preprint platforms provided details about article screening (basic checks) and 14 (32%) of these actively involved researchers with context expertise in the screening process. Almost all preprint platforms allow submission to any peer-reviewed journal following publication, have a preservation plan for read access and most have a policy regarding reasons for retraction and the sustainability of the service.

**Conclusion** A large number of preprint platforms exist for use in biomedical and medical sciences, all of which offer researchers an opportunity to rapidly disseminate their research findings onto an open-access public server, subject to scope and eligibility.

## INTRODUCTION

A preprint is an non-peer-reviewed scientific manuscript that authors can upload to a public preprint platform and make available almost immediately without formal external peer review. Posting a preprint enables researchers

## Strengths and limitations of this study

► We developed robust methodology for systematically identifying relevant preprint platforms and involved platform owners/representatives wherever possible to verify data.
► We undertook an internal pilot of developing and testing out the data collection form in collaboration with a preprint platform owner and funders.
► For platforms that had a partner journal and without verification, it was sometimes unclear if the policy information related to the journal, preprint platform or both.
► We provide a searchable database as a valuable resource for researchers, funders and policy-makers in the biomedical and medical science field to determine which preprint platforms are relevant to their research scope and which have the functionality and policies that they value most.
► We plan to update this searchable database periodically to include any new relevant preprint platforms and to amend any changes in policy.

to 'claim' priority of discovery of a research finding; this can be particularly useful for early-career researchers in a highly competitive research environment. Some preprint platforms provide digital object identifier (DOIs) for each included manuscript. This information can be included in grant applications. Indeed, progressive granting agencies are recommending applicants include preprints in their applications (eg, National Institutes of Health (NIH, USA)[1] and in the UK, preprints are becoming recognised as eligible outputs in the Research Excellence Framework exercise which assesses institutional research performance.[2]

Preprints have been widely used in the physical sciences since the early 1990s, and with the creation of the repository of electronic articles, arXiv, over 1.6 million preprints or accepted/published manuscripts have been deposited on this platform alone.[3] Since September 2003, arXiv has supported the

sharing of quantitative biology preprints under the q-bio category. The use of preprints in biomedical sciences is increasing, leading to the formation of the scientist-driven initiative Accelerating Science and Publication in biology (ASAPbio) to promote their use.[4] A preprint platform dedicated to life science-related research (bioRxiv) founded in 2013 has already attracted nearly 80 000 preprints.[5] This platform was set up to capture science manuscripts from all areas of biology, however, medRxiv was launched in June 2019 to provide a dedicated platform and processes for preprints in medicine and health related sciences[6] and it already hosts over 3400 preprints, becoming particularly popular with COVID-19. The Center for Open Science (COS)[7] has also developed web infrastructure for these new 'Rxiv' (pronounced 'archive') services,[8] while F1000 Research has provided instances of its postpublication peer review and publishing platform for use by several funders (eg, Wellcome Trust) and research institutions to encourage preprint-first research publishing.[9] Recently, several large publishers (Springer Nature, Wiley, Elsevier) have developed, codeveloped or acquired preprint platforms or services, and in April 2020, SciELO launched a preprint platform that works with Open Journal Systems.[10] Many other preprint platforms also support dissemination of biomedical and medical sciences within their broader multidisciplinary platforms.

Given the increase in the use and profile of preprint platforms, it is increasingly important to identify how many such platforms exist and to understand how they operate in relation to policies and practices important for dissemination. With this aim in mind, we conduct a review to identify all preprint platforms that have biomedical and medical science scope and contrast them in terms of their unique characteristics (eg, scope of the preprint, preprint ownership) and policies (eg, administrative checking, copyright and licensing). We also provide a searchable repository of the platforms identified so that researchers, funders and policymakers have access to a structured approach for identifying preprint platforms that are relevant to their research area.

## TERMINOLOGY

We define a preprint according to the Committee of Publication Ethics definition:

> 'A preprint is a scholarly manuscript posted by the author(s) in an openly accessible platform, usually before or in parallel with the peer review process'.[11]

Any platform or server that hosts a collection of preprints will be referred to as a preprint platform. We use 'platform' instead of 'server' because, within this definition, we include both servers with no dedicated formal peer-review service and platforms where a manuscript has been submitted for peer review and is openly available to view before the peer review is formally complete.

## METHODS
### Preprint platform identification

A preliminary list of potentially relevant preprint platforms was identified using Martyn Rittman's original list[12] and extended using a basic Google web search using the search term 'preprint' and the knowledge of the Steering Group (study authors). Additional preprint platforms that were launched up to 25 June 2019 were included.

### Preprint platform selection

We included any preprint platform that has biomedical or medical scope according to MEDLINE's journal selection criteria.[13] Generally this covers: '(manuscripts) predominantly devoted to reporting original investigations in the biomedical and health sciences, including research in the basic sciences; clinical trials of therapeutic agents; effectiveness of diagnostic or therapeutic techniques; or studies relating to the behavioural, epidemiological or educational aspects of medicine.'

We aimed to be overinclusive such that preprint platforms that hosted work within the above MEDLINE definition of scope among a broader scope (such as 'all physical, chemical and life sciences') were included. For inclusion, the platforms primary focus needed to act as a preprint platform rather than a more general repository where preprints might be incidentally deposited. Platforms were included without any language restrictions on the content accepted for posting. Eligibility of preprint platforms was arrived at in discussion with two authors (JJK and NCP) and independently approved by the Steering Group. Preprint platforms were excluded for any of the following reasons: they were no longer active (as of 25 June 2019); they were print only or had no web-presence ('offline'); their primary function was classed as a general purpose repository with no exclusive preprint functionality. We also excluded service platforms that only host postprints (after peer review), such as Science magazine's 'first release'.

### Data extraction items

A data collection form was developed by the Steering Group which aimed to capture both preprint platform characteristics and policies. The form was pilot tested with bioRxiv and revised accordingly following discussion with the platform owner. The final agreed data collection form is available online.[14] In brief, we extracted information on the preprints scope, ownership, governance, indexing and citation identifiers, submission formatting, visibility/versioning, article processing charges, publication timings, editorial board membership and for-profit or not-for-profit status. We also collected data on any checking/screening before preprint posting, open access/copyright and licensing options, sustainability and long-term preservation strategies, usage metrics and the simultaneous deposition policy relating to a manuscript submitted to a journal and the manuscript on the platform, and, if appropriate, policies about the deposition of accepted and published papers onto the platform.

## Data extraction process

Manual extraction was completed for each platform using information found on the platform's website where content was directly accessible or found on associated webpages provided by the platform (eg, the 'About' pages for many Open Science Framework, OSF, platforms linked to external websites provided by the platform operators). Verbatim text from the online search was recorded alongside any relevant web links. The completed data extraction form was then sent to the platform contacts (usually the platform owner), who were asked to check the data for completeness, fill in any missing fields and respond to any queries. Where an independent review could not be undertaken due to language barriers on the platforms website, the platform owner/representative provided the data. On receiving the responses from the platforms, the researcher updated the data form, in some cases simplifying the text records into categorised information. These data were then returned to the platform to confirm the data were accurate and as complete as possible, and these records were then recorded as 'verified by the platform representative/owner'.

If no contact with the platform was established, a second researcher independently completed the data extraction using information found on the platform's website and consensus was reached. The completed data form was sent to the platforms informing them that the included information would be presented about their platform as 'unverified' data. The deadline for preprint platforms to approve any information and to confirm that all data could be shared publicly was 19 January 2020. Further datasets and records were updated with information provided up to 27 January 2020, and are available on the Zenodo repository.[14]

## Reporting of results

The preprint platform characteristics and policies were summarised descriptively and divided into preprint platforms (1) hosted on the OSF Preprints infrastructure, (2) provided by the Open Research Central infrastructure and (3) all other eligible platforms. Characteristics are presented as: (A) the scope and ownership of each platform; (B) content-specific characteristics and information relating to submission, journal transfer options and external discoverability; (C) screening, moderation and permanence of content; (D) usage metrics and other features and (E) metadata.

## Patient and public involvement

No patients were involved in setting the research question nor were they involved in the design, implementation and reporting of the study. There are no plans to involve patients in the dissemination of results.

## RESULTS

From all sources, 90 potentially eligible preprint platforms were identified for this review, although 46 were

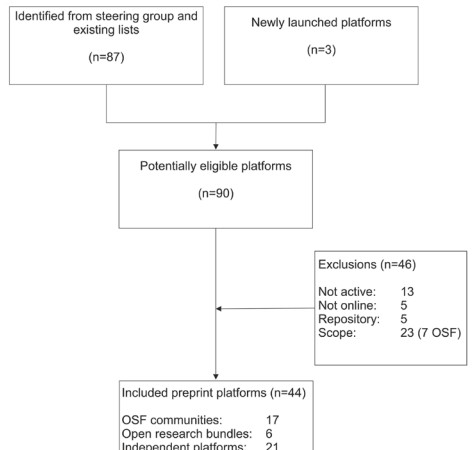

**Figure 1** Flow diagram of included preprint platforms covering biomedical and medical scope. OSF, Open Science Framework.

excluded based on scope (n=23), inactivity (n=13), no online presence (n=5) or were general repositories (n=5) (figure 1). A list of excluded preprint platforms can be found in online supplemental table 1. Of the 44 included preprint platforms, 17 were hosted by the OSF preprint infrastructure (although MarXiv is no longer part of the OSF family), 6 were provided by the Open Research Central infrastructure and 21 were other independent preprint platforms (figure 1). Of the 21 independent preprints platforms, four were First Look platforms (Cell Press Sneak Peek, Preprints with the Lancet, NeuroImage: Clinical and Surgery Open Science). While meeting the criteria for inclusion in this review, PeerJ Preprints decided to accept no new preprints after 30 September 2019. Thirty-eight (86%) of the 44 preprint platforms verified their own data. We present the data tables in this manuscript, though all tables and raw data are available in the Zenodo repository.[14] A searchable database of all the preprint platform information is also available (https://asapbio.org/preprint-servers).

## Scope and ownership of preprint platforms

Twenty-eight platforms (64%) are owned by non-profit academic groups, scientific societies or funding organisations while two platforms are owned by individuals or small communities (Frenxiv and ViXra) (online supplemental table 2). Fourteen preprint platforms (32%) are affiliated or partly owned by for-profit publishers or companies; however, the preprint service part of their operation was declared as non-profit for three of these ( Preprints.org, ESSOAr and MitoFit Preprint Archives). Of the preprint platforms associated with 'for-profit' status, only F1000 Research requires authors to pay an article processing charge.

Twenty-four (55%) preprint platforms accepted articles that covered multidisciplinary scope while 20 (45%) were discipline specific (eg, PsyArXiv for psychological research) (online supplemental table 2). Despite the multidisciplinary scope, there were some further

restrictions for some of the platforms, for example, there are five regional platforms (AfricArxiv, Arabixiv, Frenxiv, INA-Rxiv, ChinaXiv) aimed mostly at research being conducted in a specific geographical region, however, the content of these articles are globally accessible. The Open Research Central platforms also only accept articles that are funded by certain funders (eg, Wellcome Open Research platform only accepts research funded by the Wellcome Trust). Some preprint platforms also only allow articles that fit the remit of their affiliated journals (eg, Cell Press Sneak Peek). Across all platforms, the median time that they have been active is 32 months (range 10 months, medRxiv to 28 years 8 months, arXiv). In that time, over 2.72 million preprints have been posted and in 2020, two platforms (Research Square and bioRxiv) have averaged more than 2500 biomedical postings per month.

### Submission, journal transfer options and external discoverability

Where the information is known, all preprint platforms support the English language, and all accept research articles (with the exception of Thesis Commons which accepts only theses) (online supplemental table 3). Some platforms also accept other languages and other article types including research presentation slides and posters. Readers can access the full content of articles from all platforms with the exception of JMIR Preprints and some of the First Look platforms (Cell Press Sneak Peek, Preprints with the Lancet and Surgery Open Science) where reader registration is required. All platforms support PDF as the main viewing option, for some platforms this can be viewed in the browser while for others it requires a download. For all platforms, authors can submit articles using either a Word doc or as a PDF, with many platforms offering authors a choice of licensing, although where authors do not get a choice, the licence required is commonly the CC-BY licence.

In general, the OSF and many of the other platforms allow authors to submit their articles to any journal although in some cases there is facilitated submission to certain journals, for example, for bioRxiv there is a host of direct transfer journal options (online supplemental table 3). Authors submitting to F1000 Research, the Open Research platforms and all First Look platforms can only submit articles to journals associated with the platform. Where the information is available, all platforms with the exception of Therapoid and ViXra are externally indexed and most are indexed on Google Scholar.

### Screening, moderation and permanence of content

Thirty-three (75%) preprint platforms provided some detail about article screening, while two (FocUS Archive and SocArxiv) do mention checks although the details of such checks are unknown (online supplemental table 4). Therapoid does not perform any screening checks but relies on a moderation process by site users following article posting and ViXra does not perform screening

checks but will retract articles in response to issues. Fourteen (32%) preprint platforms that perform screening checks actively involved researchers with content expertise in this process. The three most common screening checks performed related to scope of the article (eg, scientific content, not spam, relevant material, language), plagiarism and legal/ethical/societal issues and compliance. Only three preprint platforms (Research Square, bioRxiv and medRxiv) check whether the content contains unfounded medical claims.

All F1000 platforms (inclusive of Open Research ones), MitoFit Preprint Archives, PeerJ Preprints and Preprints. org describe policies online in relation to NIH guidance for reporting preprints[15] with regards to plagiarism, competing interests, misconduct and all other hallmarks of reputable scholarly publishing (online supplemental table 4). Some preprint platforms do have policies but fall short of transparently making these policies visible online while some platforms have no policies. If content is withdrawn, some platforms ensure that the article retains a web presence (eg, basic information on a tombstone page) although this was not standard across all platforms. Almost all platforms have a preservation plan (or are about to implement) for read access. Most commonly, platforms have set up an archiving agreement with Portico. Others have made their own arrangements: as a notable example, the OSF platforms are associated with a preservation fund provided by the COS to maintain read access for over 50 years. In addition, most platforms have details on the sustainability of the service, for the OSF platforms this come from an external source (eg, grants to support the COS framework), while for the Open Research Central infrastructure platforms this comes from article processing charges covered by the respective funding agencies. For some of the other platforms, funding is received from either internal or external sources or from other business model services (eg, from associated journal publishing).

### Usage metrics and other features

With the exception of arXiv and MitoFit Preprint Archives (Therapoid metrics arriving soon), all preprint platforms have some form of usage metrics, and apart from JMIR Preprints and ViXra all provide the number of article downloads on the abstract page (online supplemental table 5). The OSF preprints are limited to downloads but the Open Research Central platforms also include the number of views, number of citations and altmetrics, while some of the independent platforms also include details of social media interactions direct from the platform (as opposed to the altmetric attention score). Most platforms (n=33; 75%) have some form of commenting and onsite search options (35; 80%), and some (mostly but not exclusively to the independent platforms) have alerts such as RSS feeds or email alerts.

## Metadata

Forty (91%) of platforms provided information on metadata and all provide the manuscript title, publication date, abstract and author names in the metadata (online supplemental table 6). Nearly all of these with the exception of SciELO Preprints provide a DOI or other manuscript identifier as well. The majority also offer subject categories (n=34) and licence information (n=26) but less than half include author affiliations (n=17) and funder acknowledgements (n=13). Eleven platforms (all six platforms under the Open Research Central infrastructure, Authorea, bioRxiv, ChemRxiv, F1000 Research, Research Square) offer full-text content, but only five include references in the metadata. Half of the platforms (n=22) offer a relational link to the journal publication (if it exists) in the metadata.

## DISCUSSION

Forty-four preprint platforms were identified that considered biomedical and medical scope. This review characterises each of these preprint platforms such that authors can make a more informed choice about which ones might be relevant to their research field. Moreover, funders can use the data from this review to compare platforms if they wish to explicitly support and/or encourage their researchers to use certain platforms.

Preprint platforms are fast evolving and despite our cut-off of 25 June 2019, we are aware of new eligible preprint platforms that have been or are about to be launched after this date, for example, Open Anthropology Research Repository[16] and Cambridge Open Engage.[17] However, the recent advancements in the number of preprint platforms in this field has meant that one platform in this review (PeerJ Preprints) ceased to accept new preprints from the end of September 2019 to focus on their peer-reviewed journal activities.[18] Through our searchable database (https://asapbio.org/preprint-servers), we will endeavour to keep this information up-to-date. More specifically the database will be maintained by ASAPbio for at least the next 2 years, and longer pending additional funding, but will be available as a CC BY resource. Our plan for maintenance is to enable preprint platforms to update their listings on demand, pending verification of publically accessible information by ASAPbio staff. We will periodically archive the database in Zenodo to preserve prior versions.

Due to the lack of formal external peer review for many platforms (with the exception of those platforms that follow the F1000 Research model), preprint platforms that include medical content have been criticised as they may lack quality which can lead to errors in methods, results and interpretation, which subsequently has the potential to harm patients.[19 20] This review has demonstrated the reality that many preprints do undergo some checks before going online, in contrast to the perception that preprints are not reviewed at all. Research Square, bioRxiv and medRxiv check specifically if there

is potential harm to the preprints' dissemination before peer review. Research Square also offers a transparent checklist to indicate the status of various quality assurance checks (not equivalent to scientific peer review) for each preprint.

Empirical evidence to support the use of editors and peer reviewers as a mechanism to ensure the quality of biomedical research is relatively weak[21 22] although other studies have rendered peer review as being potentially useful.[23 24] This review provides some justification that preprint platforms might be a reasonable option for researchers, especially given the time spent and associated cost of peer review.[25] In a recent survey of authors that have published with F1000 Research, 70% of respondents found the speed of publication to be important or very important.[26] In some scenarios, the time to deliver research findings may be as equally as important as research quality, and may be critical to healthcare provision. A good example of this is the current outbreak of novel coronavirus, where much of the preliminary evidence has been made available through preprints at the time of WHO declaring the epidemic a public health emergency.[27] The issue of preprints being available before peer review, and also the level of screening before a preprint is posted, has been particularly pertinent in this case. As an example, bioRxiv has rapidly adapted to ensure users appreciate there has not been any peer review of the COVID-19-related work presented on this platform. In light of COVID-19, people including the patients and the public might be interested in a quick and easy way to search across platforms. As a start at improving discoverability, Europe PMC aggregates preprints from several repositories and already nearly 3000 preprint articles with 'COVID-19' in the title are listed.[28]

### Strengths and limitations of the study

The strength of this study is that we developed robust methodology for systematically identifying relevant preprint platforms and involved platform owners/representatives wherever possible to verify data that was either unclear or not available on platform websites, and when this was not possible, a second researcher was involved in the data acquisition process. Systematically identifying web-based data that is not indexed in an academic bibliographic database is challenging,[29] though the methods employed here are compatible with the principles of a systematic search: the methods are transparent and reproducible. This approach builds on an earlier list of preprint servers,[12] the process behind which did not use systematic methods or involve platform owners as far as we are aware.

We undertook an internal pilot of developing and testing out the data collection form in collaboration with a preprint platform owner and ASAPbio staff and funders (promoters of preprint use) in order to ensure that the list of characteristics collected was both complete and relevant to different stakeholder groups including academics and funders. Many of the general policy information for some platforms was not well reported or easy to find online and therefore an unexpected but positive by-product of this research is that several of these platforms have updated their webpages to improve

the visibility and transparency of their policies in response to this research. Similarly, some platforms became aware of policy attributes that they had not previously considered and are now in the process of considering these for future implementation.

One limitation is that we focused our attention on the 'main' preprint article although in some cases different policies existed for the supplementary material, for example, acceptable formats and licensing options. This level of detail will be included in our searchable database. Another potential shortcoming was that some preprint platforms had a partner journal and without verification it was sometimes unclear if the policy information related to the journal, preprint platform or both. Finally, we defined preprint platforms as hosting work before peer review is formally complete and we acknowledge that some platforms included here also host content that has already been peer-reviewed and/or published in a journal (eg, postprints)[30]; this is unlikely to affect the interpretation of policies for preprinted works discussed herein.

### Implications for authors of biomedical and medical research

With the increase in the number of preprint platforms available in the biomedical and medical research field, authors have the option to make publicly available and gain some early ownership of their research findings with little or no cost to themselves. Moreover, with many preprints platforms there is little restriction with regard to authors later publishing their preprints in peer-reviewed journals of their choice. While we did not tabulate information on this specifically, it was noted that some platforms (notably OSF platforms) did recommend that authors check the SHERPA/RoMEO service for details of a journal's sharing policy. There is also some evidence that preprinting an article first may even boost citation rates[31] due to increased attention from tweets, blogs and news articles than those articles published without a preprint. With many platforms carrying out suitable quality-control checks and having long-term preservation strategies, preprint platforms offer authors direct control of the dissemination of their research in a rapid, free and open environment. As well as primary research, preprints are also vital to users of research (systematic reviewers and decision makers). As an example, a living mapping systematic review of ongoing research into COVID-19 is currently being undertaken, and almost all included studies to date have been identified through preprint platforms.[32]

### Implications for preprint platforms

There has been a sharp rise in the number of preprints being published each month and it has been estimated (as of June 2019), preprints in biology represents approximately 2.4% of all biomedical publications[33]; and as of April 2020 there are already over 2.72 million preprints in the platforms that we evaluated. This review has summarised the key characteristics and policies of preprint platforms posting both medical and biomedical content although there is a need for some of these platforms to update their policies and to make them more transparent online. As preprints are not formally reviewed

for scientific rigour through peer review, it is important to make it clear that their validity is less certain than for peer-reviewed articles (although even the latter may still not be valid). There is perhaps a growing need to standardise the checking process across platforms; such a process should not diminish the speed of publication (what authors value most about a preprint[22]). There is the temptation of making the checking process more rigorous, for example, by including relevant researchers within the field as gatekeepers. However, this may slow down the process of making scientific work rapidly available and may promote groupthink, blocking innovative contrarian ideas to be circulated for public open review in the preprint platforms. Based on current checks, our review shows that most preprint platforms manage to post preprints within 48 hours and all within a week on average. Further challenges may arise on resources if the number of preprints continue to rise at a similar rate and the number of new platforms begins to plateau. And now, as several initiatives progress with work to build scientific review directly onto preprints (eg, Peer Community in,[34] Review Commons,[35] PREreview[36]), it may become even more important to provide clarity about the level of checks a manuscript has already received and would need to receive to be considered 'certified' by the scientific community. If anything, the wide public availability of preprints allows for far more extensive review by many reviewers, as opposed to the typical journal peer-review where only a couple of reviewers are involved. Our review identified 14 platforms linked to for-profit publishers and companies but only F1000 Research currently charges a small article processing charge to authors. With the increase in demand and resources needed to maintain preprint platforms, we should be mindful that article processing charges may change downstream meaning that platforms may have to charge authors.

### Conclusion

One outcome of this review has been to understand the various drivers behind the proliferation of preprint platforms for the life and biomedical sciences. While arXiv, bioRxiv, chemRxiv and medRxiv aim to provide dedicated servers for academics within each field they are dedicated to, several academic groups have offered alternative subject-specific or regional services in line with their own community's needs, such as sharing work in languages other than English, using the OSF infrastructure. A third provider of preprint platforms is industry stakeholders: as academic publishers providing or acquiring preprint services to support the content they receive as submissions to their journals, and as biotechnology or pharmaceutical companies looking to support the sharing of relevant research content. Whether any platform becomes dominant may be influenced by the communities who adopt them, the influencers who promote them (funders and researchers who influence hiring and promotion decisions) and the financial sustainability underpinning them. We hope that enabling transparency into the processes and policies at each platform empowers the research community (including researchers, funders and

others involved in the enterprise) to identify and support the platform(s) that help them to share research results most effectively.

**ORCID iDs**
Jamie J Kirkham http://orcid.org/0000-0003-2579-9325
Naomi C Penfold http://orcid.org/0000-0003-0568-1194
David Moher http://orcid.org/0000-0003-2434-4206

**Author affiliations**
[1] Centre for Biostatistics, Manchester Academic Health Science Centre, University of Manchester, Manchester, UK
[2] ASAPbio, San Francisco, California, USA
[3] Murphy Mitchell Consulting Ltd, Chichester, UK
[4] Université de Paris, Centre of Research in Epidemiology and Statistics (CRESS), Inserm, Paris, France
[5] Meta-Research Innovation Center at Stanford (METRICS) and Departments of Medicine, of Epidemiology and Population Health, of Biomedical Data Science, and of Statistics, Stanford University, Stanford, California, USA
[6] Centre for Journalology, Clinical Epidemiology Program, Ottawa Hospital Research Institute, Ottawa, Ontario, Canada

**Acknowledgements** We thank John Inglis for his advice on developing the data collection form and helpful comments on the manuscript. We also thank Robert Kiley, Geraldine Clement-Stoneham, Michael Parkin, Amy Riegelman, and Claire Yang for helpful feedback and conversations. We also would like to thank collectively the preprint platform owners and representatives who provided both data and verified information.

**Contributors** JJK and DM jointly conceived the study and are the guarantors. JJK, DM and NCP designed the study methods and developed the data collection form. JJK, NCP, FM, IB, JPI, JP and DM were involved in identifying eligible platforms. JJK, NCP and FM were all involved in data extraction and JJK and NCP did the analysis and prepared the data tables. JP developed the online searchable database. JJK prepared the initial manuscript. JJK, NCP, FM, IB, JPI, JP and DM were involved in the revision of this manuscript. JJK, NCP, FM, IB, JPI, JP and DM read and approved the final manuscript and are accountable for all aspects of the work, including the accuracy and integrity.

**Funding** NCP, FM and JP received funding for ASAPbio preprint research from The Wellcome Trust, Chan Zuckerberg Initiative, Howard Hughes Medical Institute, Simons Foundation, Medical Research Council and Canadian Institutes of Health Research.

**Competing interests** JP is executive director of ASAPbio.

**Patient consent for publication** Not required.

**Ethics approval** Not required. This is a descriptive study of publicly available information made available on websites. Data was confirmed by preprint platform owners/representatives using only email contacts available on those public websites.

**Provenance and peer review** Not commissioned; externally peer reviewed.

**Data availability statement** Data are available in a public, open access repository. The data from this study are available in Zenodo (https://zenodo.org/record/3700874), which we will update periodically with a new version number as new platforms come online and policies of platforms currently identified change.

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
