## [Reviewer comments · BMJ Open]

ARTICLE DETAILS

TITLE (PROVISIONAL)	A systematic examination of preprint platforms for use in the medical and biomedical sciences setting
AUTHORS	Kirkham, Jamie J.; Penfold, Naomi C; Murphy, Fiona; Boutron, Isabelle; Ioannidis, John P; Polka, Jessica; Moher, David

VERSION 1 – REVIEW

REVIEWER	Armen Yuri Gasparyan Departments of Rheumatology and Research and Development, Dudley Group NHS Foundation Trust (Teaching Trust of the University of Birmingham, UK)
REVIEW RETURNED	02-Jul-2020

GENERAL COMMENTS	This is a descriptive study of preprint platforms. The results may help prospective authors reposit their research and reviews for setting priority. Comments. 1. It would be helpful to highlight in Methodology that only English language preprint platforms were analyzed. The Discussion may benefit from adding a line about prospects of launching non-English platforms in China, Russia and other non-Anglophone countries.2. The Discussion part may include a line about membership of preprint platforms in the Committee on Publication Ethics and adherence to ethics norms.3. Preprints are part of the Open Access Initiative. Such a point could be discussed in view of positive and negative (predatory publishing) sides of the global initiative.4. Tables can be archived as supplementary materials. 2. Tables can be not related
---

REVIEWER	Bernd Pulverer EMBO Press Germany EMBO Press, which runs the Review Commons platform in partnership with ASAPbio. bioRxiv advisory board.
REVIEW RETURNED	10-Aug-2020

GENERAL COMMENTS	General comments This analytical review provides an important, fairly detailed summary of preprint platforms/services available at this time. The article appears to be unique in providing a holistic overview of preprint and preprint-like services ranging from biological to medical sciences – it is therefore valuable and timely. The scope is broader than claimed, extending beyond the biomedical/medical sciences - the information will be relevant to all life science practitioners, funders and research institutions. The authors state that the searchable tables that
--

summarize several key attributes of preprints will be regularly updated, so that this article is likely to mark the provision of a stable, authoritative central resource for preprint services.

Publication is recommended subject to minor revision; extension of the database is encouraged.

Given the resource value of this database, it will be important to provide key attributes of this resource (see below).

The article is sometimes written in an informal style and it occasionally includes informal claims for preprint use and against peer review/journal value (e.g. p.12, l. 50-60; p. 14, l.33-34; p. 15, l. 29-33). If such statements are to be included, they have to be supported by compelling empirical evidence. Straightforward textual revision would provide for a more dispassionate, scholarly framing.

Detailed comments

- The main value of this work is as a community resource. As such, please add a more detailed description about sustainability of the database, who curates and has editorial control, frequency of updates and if there is a correction mechanism.
- The authors have chosen a rather broad definition for 'preprint platform'. That is OK, as long as there is a clear differentiation between different categories. One crucial difference is between 'preview' and 'preprint'. The former is tied to peer review or publication in a specific journal (e.g. F1000Research and Cell's 'sneak peek'). It is recommended to tabulate these separately. The difference is crucial, in particular for the authors of journal articles. It is also unclear why a service like Science magazine's 'first release' [<https://www.sciencemag.org/about-first-release>] was excluded – one assumes because it is on the AAV (post review) version; this should be more clearly stated as an exclusion criterion; note that AAV is a compliance route for funder mandates including PlanS and is thus likely to be more broadly adopted by many journals; the authors may wish to reconsider exclusion. On the other hand, it is unclear why less structured platforms such as Zenodo (<https://zenodo.org>) or Figshare (<https://figshare.com>) were excluded. These platforms can and do host biomedical preprints (defined as a scholarly publication close to the format of a research paper) alongside other formats. Suppl. Table1 lists 2 of these as excluded because they are 'Repositories' (see also p. 7, lines 15-18). It is unclear to me what the formal distinction is to platforms included in this study, beyond a more heterogeneous content (which is apparently not an exclusion criterion: p. 10, line 5-6).
- The justification for the exclusion of geographically restricted services (p. 9, line41) seems somewhat arbitrary, as the content is, I believe, globally accessible.
- A COPE definition is employed for preprint, which states '....usually before...peer review process'. This is in my view a slightly narrow definition as preprint can and should stand independent of journal publication.
- Some of the preprint platforms listed do not appear to use stable digital identifiers such as DOIs. Without this, the service is arguably not a preprint and inclusion questionable.
- It would add tremendous value to add more granularity to the database (i.e. beyond the general description in the text): 1) what type of quality control screening is applied (data integrity, plagiarism, ethics, scope, primary research vs. meta-analysis vs. review, language, author approval/affiliations/competing interests) and what process (e.g. volunteer; automated); 2) peer review; commenting (moderated?); 3) retraction/correction process; 4) preservation & sustainability (journals charges, author charges, funded); 5)

	transparent description of policies; 6) launch date/volume of preprints posted (total/biomedical sciences)  • It is reasonable to exclude inactive or terminated preprint services (suppl. Table 1). It is therefore unclear why PubPeer was prominently discussed throughout (e.g. p.9, p.12) the authors should stick to their own definition for exclusions. Minor:  • It is unclear to me if BMJ Open Journal style allows formal references to websites (with access date e.g. [12]). • p.6, l.6: change 'scientific publishing' to 'dissemination' as the former is journal specific. • p.6, l. 9: insert 'some key' characteristics and policies • p.6, l. 41-50 & p. 7, line 27: 'revised following discussion...': move Amy Riegelman and John Inglis contributions from main text to acknowledgements. Delete 'containing.....preprints' and 'if we became.....owners'. • p.7, l.18: replace 'specific' with 'exclusive'. The term 'functionality' is strange – what specific functionality do preprint only servers have? See also exclusion criteria point above. • p. 7 l. 29: delete 'in Zenodo rep.'; line 47: change 'single res...' to 'manual extraction' • p. 8, l.7 &9: delete [names] and 'on the final.....discussion' • p. 9, l.4: is 'first look' an accepted term of art? I suggested 'preview' above. • p.10, l.29: remove 'commonly' • p.10, l. 52: 'dangerous to human health' is vague: 'dual use', 'unfounded medical claims'? • p. 11, l. 14-17: delete 'for OSF.....processing'; l. 29: 'T.metrics arriving soon' • p. 12, l. 35: '..lack of formal peer review': this is incorrect as a number of services included, such as F1000R does have formal peer review on all posted content. • p.12, l. 40: 'sensible': subjective: define or delete. • p. 13l. 6: SARSCoV-2; l/ l. 38: remove 'on...produce' • p. 15, l. 6-7: peer review never 'validates' • p. 15, l. 40: OS is not the same as free science – unclear why sustainable models are regarded as 'unfortunate' • p. 16, l. 17: just list names or add affiliations for all names.
--	---

VERSION 1 – AUTHOR RESPONSE

Reviewers' Comments to Author:

Reviewer: 1

Reviewer Name: Armen Yuri Gasparyan

Institution and Country: Departments of Rheumatology and Research and Development, Dudley Group NHS Foundation Trust (Teaching Trust of the University of Birmingham, UK) Please state any competing interests or state 'None declared': None

This is a descriptive study of preprint platforms. The results may help prospective authors reposit their research and reviews for setting priority. Comments.

1. It would be helpful to highlight in Methodology that only English language preprint platforms were analyzed.

RESPONSE: we did not limit our preprint platforms to English language only – this is evident in our Table 2 which outlines content languages accepted. As an example, for the preprint platform 'Arabixiv' – the main language is Arabic even though English is also accepted.

Text updated to suggest platforms were included without any language restriction on the content posted. However in some cases where the website was not in English, we were unable to do an independent review and the server owners/representatives provided their own data. Text also amended.

The Discussion may benefit from adding a line about prospects of launching non-English platforms in China, Russia and other non-Anglophone countries.

RESPONSE: see point above. The purpose of this review is to summarise the current practices and policies of those preprint platforms that currently exist and not to recommend that more platforms are needed.

2. The Discussion part may include a line about membership of preprint platforms in the Committee on Publication Ethics and adherence to ethics norms.

RESPONSE: Thanks for this comment. COPE membership guidelines (<https://publicationethics.org/resources/guidelines-new/principles-transparency-and-bestpractice-scholarly-publishing>) imply that only journals may join; presently we are not aware that the idea of offering COPE membership to preprint servers mentioned in the 2018 discussion document we have cited has come to fruition.

3. Preprints are part of the Open Access Initiative. Such a point could be discussed in view of positive and negative (predatory publishing) sides of the global initiative.

RESPONSE: Again thanks for this comment. It is very unlikely that preprints would be of any interest to predatory journals or publishers. Preprint servers do not charge a fee for articles that are successfully screened for inclusion on a preprint server. Predatory journals are almost entirely focused on nefarious ways of obtaining a fee in the form of an article processing charge (APC). A preprint server seems counter intuitive to their desire to generate income.

4. Tables can be archived as supplementary materials.

RESPONSE: Thanks for this comment. We have raised this point with the editors and we are happy to include tables in accordance to their recommendations

Reviewer: 2

Reviewer Name: Bernd Pulverer

Institution and Country: EMBO Press, Germany Please state any competing interests or state 'None declared': EMBO Press, which runs the Review Commons platform in partnership with ASAPbio. bioRxiv advisory board.

General comments

This analytical review provides an important, fairly detailed summary of preprint platforms/services available at this time. The article appears to be unique in providing a holistic overview of preprint

and preprint-like services ranging from biological to medical sciences – it is therefore valuable and timely. The scope is broader than claimed, extending beyond the biomedical/medical sciences - the information will be relevant to all life science practitioners, funders and research institutions. The authors state that the searchable tables that summarize several key attributes of preprints will be regularly updated, so that this article is likely to mark the provision of a stable, authoritative central resource for preprint services.

Publication is recommended subject to minor revision; extension of the database is encouraged. Given the resource value of this database, it will be important to provide key attributes of this resource (see below).

RESPONSE: Thanks for this positive comment. We outline some plans below to update and extend the database in light of this reviewers recommendations, however, many of these additional requests were outside the scope of this initial manuscript version of the review.

The article is sometimes written in an informal style and it occasionally includes informal claims for preprint use and against peer review/journal value (e.g. p.12, l. 50-60; p. 14, l.33-34; p. 15, l. 29-33). If such statements are to be included, they have to be supported by compelling empirical evidence. Straightforward textual revision would provide for a more dispassionate, scholarly framing.

RESPONSE – Thanks for this comment. We will leave this to the editor’s discretion. In the sections questioned, we do provide references to where the information was found albeit these sources may not result from ‘compelling empirical evidence’. Compelling empirical evidence in this field is sparse at best and we believe these insights we include provide a bit of added flavour for readers to consider themselves – otherwise this review would be a quite a dry read.

Detailed comments

- The main value of this work is as a community resource. As such, please add a more detailed description about sustainability of the database, who curates and has editorial control, frequency of updates and if there is a correction mechanism.

RESPONSE: The database will be maintained by ASAPbio for at least the next two years, and longer pending additional funding, but it is available as a CC BY resource so can be forked if needed.

Our plan for maintenance is to enable the preprint servers to update their listings on demand, pending verification of publicly-accessible information by ASAPbio staff.

We will periodically archive the database in Zenodo to preserve prior versions. Future manuscript updates will be driven by substantial changes, e.g. major policy overhauls or a substantial number of new platforms going online. The sustainability plan is added to the discussion

- The authors have chosen a rather broad definition for ‘preprint platform’. That is OK, as long as there is a clear differentiation between different categories. One crucial difference is between ‘preview’ and ‘preprint’. The former is tied to peer review or publication in a

specific journal (e.g. F1000Research and Cell’s ‘sneak peek’). It is recommended to tabulate these separately. The difference is crucial, in particular for the authors of journal articles.

RESPONSE: There are many ways to structure this review according to individuals preference, the agreed approach by our Steering group was by platform type a) OSF communities, b) Open Research Central infrastructure and c) other platforms. The details about platforms tied to peer reviewed journals of course appears in our details in Table 2. However, we agree with your point - "preview" does add more information about the review status of the preprint, but there are cases where manuscripts get rejected and become mere "preprints" again. For the database version we will consider expressing this field as "Does preprint posting require journal submission?" and we could add that as a field in the database revision for the website.

It is also unclear why a service like Science magazine's 'first release' [<https://www.sciencemag.org/about-first-release>] was excluded – one assumes because it is on the AAV (post review) version; this should be more clearly stated as an exclusion criterion; note that AAV is a compliance route for funder mandates including PlanS and is thus likely be more broadly adopted by many journals; the authors may wish to reconsider exclusion.

RESPONSE: First Release hosts only postprints (after peer review) and does not allow authors to post preprints. We see compliance with Plan S and open access mandates as separate from preprints; while Coalition S encourages their posting, Plan S covers peer reviewed manuscripts: <https://www.coalition-s.org/addendum-to-the-coalition-s-guidanceon-the-implementation-of-plan-s/principles-and-implementation/>. Exclusion criteria updated in manuscript to suggest were are not including postprints (after peer review).

On the other hand, it is unclear why less structured platforms such as Zenodo (<https://zenodo.org>) or Figshare (<https://figshare.com>) were excluded. These platforms can and do host biomedical preprints (defined as a scholarly publication close to the format of a research paper) alongside other formats. Suppl. Table1 lists 2 of these as excluded because they are 'Repositories' (see also p. 7, lines 15-18). It is unclear to me what the formal distinction is to platforms included in this study, beyond a more heterogenous content (which is apparently not an exclusion criterium: p. 10, line 5-6).

RESPONSE: While some preprint platforms also host postprints, our criteria were focused on platforms that aim first and foremost to act as a preprint server rather than as a more general repository where preprints might incidentally be deposited – this is the case for those platforms mentioned by this reviewer. Manuscript exclusion criteria updated. Incidentally we corresponded with Zenodo and they preferred not to be referred to as a preprint platform.

- The justification for the exclusion of geographically restricted services (p. 9, line41) seems somewhat arbitrary, as the content is, I believe, globally accessible.

RESPONSE: We did not exclude these geographically restricted services- their data appears in our tables. On p9, line 41, we simply state that regional platforms exist. Text amended to make it clear that the article content is still globally accessible.

- A COPE definition is employed for preprint, which states '...usually before...peer review process'. This is in my view a slightly narrow definition as preprint can and should stand independent of journal publication.

RESPONSE: Thanks for your comment. The steering group explored a number of preprint definitions that are citable from credible organisations. On the contrary we felt that the

COPE definition was broad enough to encompass the scope of our review. Many other definitions include the phrase 'without peer review'. Our scope was to include those platforms that allow post-publication peer review such as F1000 Research. We felt this was an important inclusion as the article acts as a preprint until the peer review aspect of the process is 'Approved'.

- Some of the preprint platforms listed do not appear to use stable digital identifiers such as DOIs. Without this, the service is arguably not a preprint and inclusion questionable.

RESPONSE: Thanks for your comment. As you point out most platforms do have a DOI but not all. All platforms included in this review have been identified as preprint platforms by their owners or our expert steering group. Furthermore, having a stable digital identifier was not part of the inclusion criteria for this review. We identify the unique identifiers for all platforms (if any) in Table 2. Included platforms without these identifiers is useful to report to potential users of preprints, should such identifiers be an important consideration for using that platform.

- It would add tremendous value to add more granularity to the database (i.e. beyond the general description in the text): 1) what type of quality control screening is applied (data integrity, plagiarism, ethics, scope, primary research vs. meta-analysis vs. review, language, author approval/affiliations/competing interests) and what process (e.g. volunteer; automated); 2) peer review; commenting (moderated?); 3) retraction/correction process; 4) preservation & sustainability (journals charges, author charges, funded); 5) transparent description of policies; 6) launch date/volume of preprints posted (total/biomedical sciences)

RESPONSE: Thanks for this comment. While we agree this information would be useful, this level of detail was beyond the scope of this initial review as we wanted to focus on summarising some of the higher level characteristics and policies of each identified platform in the first instance. Furthermore, this level of detail and the visibility of this more 'granular' detail was quite sparse – this is something many preprint owners are looking to address following this research - a comment that is made in the manuscript discussion.

If any extra details were provided on specific aspects, we have included the weblinks to this information in the Zenodo files. In line with our response to the updating of the database above, we will ensure in the future that any additional important information is included that was not specifically captured in this review.

- It is reasonable to exclude inactive or terminated preprint services (suppl. Table 1). It is therefore unclear why PubPeer was prominently discussed throughout (e.g. p.9, p.12) the authors should stick to their own definition for exclusions.

RESPONSE: We assume this reviewer is referring to Peer J Preprints and not PubPeer here. We excluded inactive or terminated preprint platforms on the basis that a) they may no longer be useful to users of preprint services because you can no longer post to them and b) we would unlikely be able to find reliable data / suitable contact about these redundant platforms.

We believe we have followed our definition of inclusion. Our cut off for accepting newly launched platforms for inclusion in the review was set to be 25th July 2019. We had to impose this date of 25th July (and stick by it) as the landscape was changing fairly quickly.

We also state on page 7 line 15 that platforms would be excluded if they were no longer active on this same date (June 25th 2019). PeerJ preprints continued to be active until 30th September 2019 which exceeds this date. We had already collected the data on this platform before the announcement about the platforms closure was made and therefore we prefer to keep in this data at this time, but acknowledge we may remove this from future updates of this review.

Minor:

- It is unclear to me if BMJ Open Journal style allows formal references to websites (with access date e.g. [12]).

RESPONSE: journal guidance has been followed

- p.6, l.6: change 'scientific publishing' to 'dissemination' as the former is journal specific.

RESPONSE: text amended

- p.6, l. 9: insert 'some key' characteristics and policies

RESPONSE: text amended

- p.6, l. 41-50 & p. 7, line 27: 'revised following discussion...': move Amy Riegelman and John Inglis contributions from main text to acknowledgements. Delete 'containing.....preprints' and 'if we became.....owners'.

RESPONSE: text amended

- p .7, l.18: replace 'specific' with 'exclusive'. The term 'functionality' is strange – what specific functionality do preprint only servers have? See also exclusion criteria point above.

RESPONSE: Text updated. As mentioned in a previous comment 'functionality' relates to the platforms unique identifier that what you are submitting is a preprint (as opposed to a dataset or postprint). This is updated in the inclusion/exclusion criteria.

- p. 7 l. 29: delete 'in Zenodo rep.'; line 47: change 'single res...' to 'manual extraction'

RESPONSE: text amended

- p. 8, l.7 &9: delete [names] and 'on the final.....discussion'

RESPONSE: text amended

- p. 9, l.4: is 'first look' an accepted term of art? I suggested 'preview' above.

RESPONSE: 'First Look' is a specific term used by these platforms. No changes made.

- p.10, l.29: remove 'commonly'

RESPONSE: text amended

- p.10, l. 52: 'dangerous to human health' is vague: 'dual use', 'unfounded medical claims'?

RESPONSE: text amended

- p. 11, l. 14-17: delete 'for OSF....processing'; l. 29: 'T.metrics arriving soon'

RESPONSE: we prefer to keep this statement as it was information currently provided by the servers. We will update any changes at the point they arrive.

- p. 12, l. 35: '..lack of formal peer review': this is incorrect as a number of services included, such as F1000R does have formal peer review on all posted content.

RESPONSE: text amended to suggest that those platforms that follow the F1000 Research Ltd model are an exception (as they offer post-publication peer review).

p.12, l. 40: 'sensible': subjective: define or delete.

RESPONSE: text amended

- p. 13l. 6: SARSCoV-2; l/ l. 38: remove 'on...produce'

RESPONSE: text amended

- p. 15, l. 6-7: peer review never 'validates'

RESPONSE: Agree – have changed to 'reviewed for scientific rigor'

- p. 15, l. 40: OS is not the same as free science – unclear why sustainable models are regarded as 'unfortunate'

RESPONSE: text deleted 'this would be unfortunate in light of the open science movement'

- p. 16, l. 17: just list names or add affiliations for all names.

RESPONSE: text amended

VERSION 2 – REVIEW

REVIEWER	Armen Yuri Gasparyan Dudley Group NHS Foundation Trust, Teaching Trust of the University of Birmingham, UK
REVIEW RETURNED	29-Aug-2020

GENERAL COMMENTS	I am satisfied with all amendments. I just wanted to clarify that preprints may include reports that cannot stand for peer review and can be withdrawn due to ethics and other concerns. COPE offers membership to editing agencies, publishers and other organizations. Preprint platforms membership in and adherence to COPE standards would improve the quality of the platforms. And focus on English preprints would be more appropriate.
---

REVIEWER	Bernd Pulverer EMBO Germany EMBO Press, which runs the Review Commons platform in
-----------------	--

	partnership with ASAPbio. bioRxiv advisory board.
REVIEW RETURNED	14-Sep-2020

GENERAL COMMENTS	Thank you for the detailed responses. The manuscript can in my view be published - it is up to the editors of the journal to consider if the revision addresses the points raised by the referees sufficiently. I highlight a couple of points to consider: 1) suggestion to tone down informal or anecdotal statements on preprint use and against peer review/journal value (e.g. p.12, l. 50-60; p. 14, l.33-34; p. 15, l. 29-33). 2) difference between 'preview' and 'preprint': in the author response the main structure note is a) OSF communities, b) Open Research Central infrastructure and c) other platforms. b) is a not-for-profit organisation based exclusively on the F1000 platform to 'formal publication on one of the open research publishing-compliant platforms.' [see https://openresearchcentral.org/about]. a) is restricted to platforms by the Center for Open Science. In my view it is debatable if a) & b) are not too specific a subdivision, and if b) is in fact a preprint platform, which is the headline topic of the article. I restate the point that the user should be clear if they are posting on a 'journal'-linked platform (even if it is pre-review) or a jnl. independent preprint platform, as this affects the options a user has for the final publication destination of a research article. It is reasonable to exclude post-peer review articles ('postprints'), but note that e.g. all F1000R articles start as 'preprints' and finish as 'postprints', so the inclusion/exclusion criteria remain inherently vague. This could be discussed more clearly. Excuse the typo in my previous report re. Peer J Preprints. The response to this point is fine.
---